# Sustainable Valorization of Tomato Pomace (*Lycopersicon esculentum*) in Animal Nutrition: A Review

**DOI:** 10.3390/ani12233294

**Published:** 2022-11-25

**Authors:** Shengyong Lu, Shengchang Chen, Haixia Li, Siwaporn Paengkoum, Nittaya Taethaisong, Weerada Meethip, Jariya Surakhunthod, Boontum Sinpru, Thakun Sroichak, Pawinee Archa, Sorasak Thongpea, Pramote Paengkoum

**Affiliations:** 1School of Animal Technology and Innovation, Institute of Agricultural Technology, Suranaree University of Technology, Muang, Nakhon Ratchasima 30000, Thailand; 2Institute of Animal Nutrition and Feed Science, Guizhou University, Guiyang 550025, China; 3Animal Nutrition and Technology Quality Control R&D Department, Guizhou Province Chuanpai Feed Co., Ltd., Guiyang 550201, China; 4Program in Agriculture, Faculty of Science and Technology, Nakhon Ratchasima Rajabhat University, Muang, Nakhon Ratchasima 30000, Thailand

**Keywords:** antioxidant, economic benefits, feed, nutritional value, tomato pomace

## Abstract

**Simple Summary:**

The global annual production of tomatoes (*Lycopersicon esculentum*) is 170 million tons. After industrial processing, a large amount of tomato pomace is produced because it contains a high amount of water and nutrients. Therefore, if it is not used properly, not only will resources be wasted, but the environment will be polluted. In addition, tomato pomace is also rich in antioxidants, such as carotenoids, lycopene, and flavonoids, which help to improve the antioxidant properties of animals. Hence, this review focuses on the nutritional content of tomato pomace and the effects of its application in livestock.

**Abstract:**

Under the background of the current shortage of feed resources, especially the shortage of protein feed, attempts to develop and utilize new feed resources are constantly being made. If the tomato pomace (TP) produced by industrial processing is used improperly, it will not only pollute the environment, but also cause feed resources to be wasted. This review summarizes the nutritional content of TP and its use and impact in animals as an animal feed supplement. Tomato pomace is a by-product of tomato processing, divided into peel, pulp, and tomato seeds, which are rich in proteins, fats, minerals, fatty acids, and amino acids, as well as antioxidant bioactive compounds, such as lycopene, beta-carotenoids, tocopherols, polyphenols, and terpenes. There are mainly two forms of feed: drying and silage. Tomato pomace can improve animal feed intake and growth performance, increase polyunsaturated fatty acids (PUFA) and PUFA n-3 content in meat, improve meat color, nutritional value, and juiciness, enhance immunity and antioxidant capacity of animals, and improve sperm quality. Lowering the rumen pH and reducing CH_4_ production in ruminants promotes the fermentation of rumen microorganisms and improves economic efficiency. Using tomato pomace instead of soybean meal as a protein supplement is a research hotspot in the animal husbandry industry, and further research should focus on the processing technology of TP and its large-scale application in feed.

## 1. Introduction

The International Feed Industry Federation (IFIF) reports that the world population will exceed 10 billion by 2050 [1]. By then, world food production will have to increase by 70% to meet the needs of human consumption [2]. In addition, the increased population will consume twice as many animal products, with pork and poultry consumption projected to grow by 105% and 173%, respectively [3,4], which will challenge the production of feed. On one hand, the growing population directly competes with animals for food [5]; on the other hand, the increase in the population’s demand for livestock products has led to the expansion of the scale of animal husbandry, thereby aggravating the shortage of feed resources [6]. Therefore, new feed resources must be found. A large number of agricultural and sideline products, food industry by-products, and insects with high protein content have gradually begun to be used by humans for the development and utilization of animal feed resources [7,8].

The annual global production of tomato (*Lycopersicon esculentum*) is 170 million tons, of which 127.5 million tons are used for fresh consumption and 42.5 million tons are used for industrial processing [9]. Industrial processing produces a large amount of TP, most of which is not properly utilized. This not only wastes resources, but also seriously pollutes the environment. Because TP is rich in water and nutrients, it is perishable and can be a contaminant if not handled properly [10]. However, if these by-products are dealt with reasonably, they can generate high economic value [11]. Tomato pomace consists of approximately 60% seeds and 40% peel, with an average protein content of 21.9% in TP and 38.7% in defatted tomato seeds [12]. The processed by-products also contain high value-added compounds, such as vitamins, carotenoids, lycopene, flavonoids, and soluble dietary fiber (SDF) [13,14,15]. Correia et al. [16] found that the addition of TP to pig diets rich in fat significantly increased the α-tocopherol content of meat and liver, and thus the oxidative stability of pork. Caluya et al. [17] showed that feeding animals 6% fresh TP significantly increased feed consumption and reduced feed cost per kilogram of body weight gain. When the tomato supplementation level was 35%, the final body weight, total weight gain, average daily weight gain, and weed intake increased significantly. Mizael et al. [18] reported that, although the addition of TP to the dairy goat diet had no difference in feed efficiency and feed conversion ratio, it improved milk quality and fat content. Mohammed et al. [19] found that the addition of 4% and 6% TP had no adverse effects on chicken growth performance, meat quality, or hormone secretion. The above results indicate that the nutrients and antioxidant properties in TP can be absorbed and digested by animals and can improve the oxidative protection ability of animals. While there have been some comprehensive evaluations of the value-added use of TP in various industries, the evaluation of the use of TP as feed has been described only marginally. Therefore, this review focuses on the nutritional value of TP and its value-added utilization in the feed industry, and provides some suggestions for the reuse of TP.

## 2. The Production and Nutritional Value of TP

### 2.1. Production of TP

The processing and uses of TP are shown in Figure 1. Tomato pomace is a by-product of tomato processing, divided into peel, pulp, and tomato seeds, accounting for 10–40% of all processed tomatoes [20]. Tomato pomace contains nearly 33% seeds, 27% peel, and 40% pulp, while dried pomace contains approximately 44% seeds and 56% peel and pulp [21]. Management of TP is considered to be an important issue for tomato processing companies. Because it is rich in nutrients and water, it easily breeds corruption, produces flies, and pollutes the environment [22,23]. Therefore, the recycling of TP is of great significance. In addition to being rich in protein, fat, minerals, fatty acids, and amino acids, TP is rich in other bioactive compounds with antioxidant properties, such as lycopene, beta-carotenoids, tocopherols, polyphenols, and terpenes [23,24]. These substances have various applications in the food [15], cosmetic [25], pharmaceutical [26], and feed industries [27,28]. This not only reduces environmental pollution and the cost of processing TP, but also eases the pressure on animal feed resources.

### 2.2. Regular Nutritional Content of TP

The nutrient levels of TP are shown in Table 1. For different places, different growth periods and different processing methods, the nutrient levels of TP will be different [29,30]. The average moisture content of TP was 73.4 g/kg, ranging from 59.6 to 88.4 g/kg. The protein content ranged from 149.5 to 298.5 g/kg, with an average of 206.5 g/kg. The fat content ranged from 85.2 to 244.7 g/kg, with an average of 128.7 g/kg. The total dietary fiber content ranged from 115.0 to 663.0 g/kg, with an average of 376.9 g/kg. These research results show that TP has less ash content and is rich in protein, fat, and total dietary fibers, which provides theoretical support for animal feed. The average nutrient composition of TP is shown in Figure 2.

### 2.3. Antioxidant Potency of TP

In addition to being rich in nutrients, TP also contains high levels of antioxidants. The antioxidant content of TP is shown in Table 2. The content of total phenol content (TPC) ranged from 94.5 to 213.4 mg GAE/g, with an average of 161.8 mg GAE/g. The total flavonoid (TFC) ranged from 30.6 to 378.7 mg QE/g, with an average of 124.4 mg QE/g. The lycopene content ranged from 36.7 to 50.2 g/kg, with an average of 44.6 g/kg. DPPH radical scavenging activity ranged from 29.9% to 75.0%, with an average of 52.5%. β-carotene bleaching inhibition activity ranged from 80.6% to 211.0%, with an average of 134.3%.

### 2.4. Mineral Composition

Tomato pomace is rich in minerals, especially Ca, P, Mg, Na, and K. The mineral content of TP is shown in Table 3. The Ca content ranged from 76.4 to 160.0 g/kg, with an average of 170.3 g/kg. There was only one result for P content, which was 219.7 g/kg. The Mg content ranged from 3.1 to 251.1 g/kg, but only one result was 3.1 g/kg. Other studies were all above 100 g/kg, with an average of 149.7 g/kg. The Na content ranged from 47.2 to 191.7 g/kg, with an average of 97.7 g/kg. The K content was the highest, ranging from 303.0 to 1125.0 g/kg, with an average of 835.5 g/kg. Fe and Zn content was minimal: Fe content ranged from 1.5 to 11.0 g/kg, with an average of 3.7 g/kg, and the Zn content ranged from 0.5 to 6.3 g/kg, with an average of 3.4 g/kg.

### 2.5. Fatty Acid Profile

The fatty acid content of TP is shown in Table 4. The most abundant saturated fatty acids were palmitic acid (C16:0), ranging from 133.9 to 205.3 g/kg, stearic acid (C18:0), ranging from 43.5 to 63.6 g/kg, arachidic acid (C20:0), ranging from 4.8 to 12.6 g/kg, behenic acid (C22:0), ranging from 1.5 to 8.2 g/kg, and tricosanoic acid (C23:0), ranging from 0.2 to 15.2 g/kg. Lignoceric acid (C24:0) levels ranged from 1.7 to 10.1 g/kg with an average level of 4.75 g/kg. However, the content of pentacosanoic acid (C25:0) and ceric acid (C26:0) was almost absent. The highest content of monounsaturated fatty acids was oleic acid (C18:1n9), ranging from 106.0 to 198.6 g/kg. The highest content of polyunsaturated fatty acids was linoleic acid (C18:2n6), ranging from 398.0 to 520.5 g/kg. SFA content ranged from 190.0 to 322.2 g/kg. Monounsaturated fatty acid (MUFA) content ranged from 110.0 to 207.9 g/kg. The n-6 PUFA content ranged from 398.6 to 530.7 g/kg. The n-3 PUFA content ranged from 42.2 to 156.6 g/kg. The MUFA/SFA ratio ranged from 34.1% to 105.2%, and the n-3 PUFA/n-6 PUFA ratio ranged from 8.0% to 39.5%.

### 2.6. Amino Acid Profile

The amino acid profile of TP is shown in Table 5. The most abundant essential amino acids were phenylalanine (average: 16.5 g/kg; from 6.1 to 50.2 g/kg), leucine (average: 16.0 g/kg; from 1.5 to 50.7 g/kg), arginine (average: 15.3 g/kg; from 1.4 to 43.4 g/kg), and lysine (average: 13.1 g/kg; from 1.7 to 44.0 g/kg). The content of histidine ranged from 0.5 to 36.4 g/kg, isoleucine was from 0.8 to 38.6 g/kg, and the content of methionine ranged from 1.2 to 10.2 g/kg. The least indispensable amino acid was threonine (average: 9.5 g/kg; from 4.3 to 23.4 g/kg) and valine (average: 12.55 g/kg; from 1.2 to 45.8 g/kg). Therefore, the content of various essential amino acids in TP was relatively balanced. The content of various indispensable amino acids can basically meet the needs of pigs and chickens.

## 3. Antioxidant Mechanisms of Bioactive Substances

Free radicals are reactive oxygen species; they are reactive chemicals with unpaired electrons, including hydrogen peroxide, hydroxyl radicals, nitric oxide, peroxynitrite, singlet oxygen, peroxyl radicals, and superoxide anions, which are ubiquitous in biological and food systems [51]. Excessive production of these reactive substances will lead to oxidative stress caused by an imbalance in the body’s antioxidant defense system and the formation of free radicals [52]. Antioxidants play a vital role in both the food system and the body, reducing the oxidative process and harmful effects of ROS, effectively inhibiting the initiation or propagation of oxidative chain reactions, thereby delaying or inhibiting the oxidation of lipids or other molecules and thus counteracting oxidative damage compounds in animal tissues [53,54,55]. In food, the formation of lipid peroxidation and secondary lipid peroxidation products can be prevented by using nutritional antioxidant molecules, which help to maintain the flavor, color, and quality of food during production and storage [56]. Sources of natural antioxidants are primarily plant phenolics, which may be present in all parts of plants, including fruits, vegetables, seeds, leaves, roots, and bark [57,58]. Plants produce numerous secondary metabolites in their normal metabolic pathways, such as flavonoids, essential oils, alkaloids, lignans, terpenes, terpenoids, tocopherols, phenolic acids, phenols, peptides, multifunctional organic acids [59,60,61], and some minerals found in nature, such as selenium and iron [62,63]. Interactions between these antioxidants may have an effect that is not a necessary property of the individual ingredients [64]. The antioxidant bioactive substances, such as phenolic compounds, carotenoids, and vitamin E, contained in TP are destined to have a promising antioxidant capacity in biological systems.

### 3.1. Total Phenol Content (TPC)

Total phenols are a group of compounds widely distributed in nature ranging from simple phenols to highly polymeric compounds, of which the presence of polyphenols and flavonoids is the main reason for their antioxidant activity [65,66,67]. Polyphenols are phytonutrients with antioxidant activity, including simple phenols, phenolic acids, coumarins, flavonoids (flavanones, flavonoids, and flavonols), as well as oligomers and polymers, such as tannins and lignin [68,69,70]. The chemical structure of these substances affects their biological activities, such as their bioavailability, absorption, interactions with cellular receptors, and enzymes [71,72]. The absorbed polyphenols are bound in the intestinal mucosa and internal tissues by methylation, sulfation, and glucuronidation or binding [73,74]. After eating foods rich in polyphenols, the antioxidant capacity of animal plasma increases, which indirectly proves that the intestinal barrier can absorb polyphenols [75,76].

There are two main methods by which a homeostasis of antioxidant balance is achieved: an enzymatic antioxidant, mainly through the triplet of catalase, superoxide dismutase, and glutathione peroxidase [77,78], and non-enzymatic antioxidants, especially low molecular weight antioxidants, such as phenols, vitamin E, and carotenoids [79,80,81]. These exogenous antioxidants are the first line of defense against antioxidant free radicals of the body [82], and flavonoids are the most effective of all polyphenols [83]. Ahmed showed that polyphenols can increase serum catalase (CAT), glutathione peroxidase (GSH-Px), and superoxide dismutase (SOD) levels, and reduce malondialdehyde (MDA) production in rats [84,85]. Polyphenols in general protect cells from free radicals by the following mechanisms: increased antioxidant enzyme activity, the inhibition of lipid peroxidation, the synergistic scavenging of free radicals with other nutrients, and the reduction of oxidation through metal ion chelation [86,87]. Together, these mechanisms protect cells from oxidative damage.

### 3.2. Total Flavonoid Content (TFC)

Total flavonoids generally refer to flavonoids. Flavonoids are a general term for a series of compounds that have 2-phenylchromone as the skeleton and are connected to each other through three carbon atoms—that is, a general term for a class of compounds with a C6-C3-C6 structure [88,89,90]. They have a variety of biological activities, including antioxidant, antibacterial, and antiviral effects, and protection against various diseases such as cancer, cardiovascular disease, and inflammation [91]. More than 6000 flavonoids have been identified to date [92], and they can be divided into six categories: flavonols, flavonoids, isoflavones, flavanones, flavanols, and anthocyanins [81,93,94,95,96,97] (the chemical structure is shown in Figure 3). In fact, they comprise a group of polyphenolic compounds produced in plants as secondary metabolites [98]. Therefore, the antioxidant properties of flavonoids are well-known [99,100]. Their antioxidant mechanism is similar to that of polyphenols. They are directly oxidized by free radicals to form less active substances through four mechanisms: (a) inhibiting nitric oxide synthase activity, (b) inhibiting xanthine oxidase activity, (c) regulating channel pathways, and (d) interacting with other enzymatic systems [101,102,103,104]. The effect on animal antioxidants will be discussed below.

### 3.3. Carotenoids

Carotenoids is a general term for a class of important natural pigments, which are commonly found in the yellow, orange-red, or red pigments of higher plants, fungi, and algae, and are fat-soluble plant pigments that constitute part of the human diet [105,106]. These compounds are capable of reacting with a wide variety of reactive species and yield numerous oxidation products with similar or even higher reactivity than their parent compounds [105,107,108,109]. There are two main classes of carotenoids: (i) highly unsaturated hydrocarbons including α-, β-, and γ-carotene and lycopene; (ii) lutein: lutein, β-cryptoxanthin, and zeaxanthin [110]. Lutein has the fewest oxygen-containing groups on its terminal rings, while unsaturated hydrocarbon carotenoids contain only carbon and hydrogen atoms and no oxygen [111]. Lycopene accounts for 80–90% of carotene [112]. It is the most effective free radical scavenger with more than twice the capacity of β-carotene [113] and 10 times that of α-tocopherol [114,115]. Lycopene has better antioxidant properties than other carotenoids, and its singlet oxygen quenching rate constant is 100 times that of vitamin E [116]. It has the ability to reduce the risk of cardiovascular disease and cancer [117,118]. Carotenoids and lutein can be arranged to form a common C40 structure; the biosynthesis of C30 and C50 carotenoids has been described in bacteria, while apocarotenoids are metabolized by shortening the C40 structure of the parent compound [119]. Carotenoids are characterized as bright yellow to red, although some colorless precursors may contribute to the common different carotenoid profiles that accumulate in organisms. Its color varies with the number of conjugated double bonds. The more the number of conjugated double bonds, the more the color shifts to red [120], which may contribute to the color brightness of animal products [121,122,123]. The bioavailability and efficacy of carotenoids depend on a number of factors: food factors, such as the fat and fiber in the food, since they are lipophilically active [124,125]; the content and availability of carotenoids in plant tissues; the locations and types of carotenoids; and the interactions between carotenoids and other compounds, such as isomeric forms of carotenoids. Therefore, although carotenoids are found in large quantities in plants, this does not necessarily mean that they will be highly bioavailable [126,127].

### 3.4. DPPH Radical Scavenging Activity

Substances capable of donating hydrogen or electrons to DPPH˙ (nitrogen-centered free radical) can be considered antioxidants; thus, DPPH can be regarded as a free radical scavenger [128]. DPPH is a stable free radical at room temperature, accepting electrons or hydrogen radicals to become stable diamagnetic molecules. It is often considered a model for lipophilic free radical activity because the molecule does not dimerize like most other free radicals because of the delocalization of spare electrons across the molecule [129,130]. Its stability mainly comes from the steric barriers of the three benzene rings that are stabilizing by resonance, such that the unpaired electrons on the nitrogen atom sandwiched in the middle cannot play their due electron pairing role [131,132]. The harmful effects of free radicals can be blocked by compounds that are antioxidant substances that can delay or inhibit the oxidation of lipids or other molecules by inhibiting the initiation or propagation of oxidative chain reactions [133], or convert free radicals into waste by-products that are excreted from the body [134]. The level of DPPH free radical scavenging is a mature mechanism for screening the antioxidant capacity of active plant substances [104]. The antioxidant capacity of TP is beyond doubt, which will be discussed later. Furthermore, in addition to the antioxidant actives mentioned in this review, TP also contains numerous actives, such as tocopherols, phytosterols, and vitamin C, of which other, more detailed actives and physiological functions have been investigated and summarized by Abbasi-Parizad [135] and Kum [136].

## 4. Nutrition of TP on Poultry

Table 6 shows that Mohammed et al. [19] fed 1–42-day-old Indian River chicks (IR) and Cobb chicks with 0%, 4%, and 6% TP, respectively, and the results showed that the TP group significantly increased the feed cost, total variable cost, and total cost. The chickens consumed more feed, but the pH of the muscles lowered. Supplementation with 6% had no effect on muscle water holding capacity (WHC) or drip loss (48 h), the mRNA expression of hepatic growth hormone receptor gene (GHR), or insulin-like growth factor-1 (IGF-1). Reda et al. [137] showed that the addition of 12% TP to a diet can significantly improve immune performance, antioxidant performance, and the digestive enzymes of Japanese quail. It reduced the cholesterol and low-density lipoprotein (LDL); increased the high-density lipoprotein (HDL), egg weight, and hatchability, the largest of which was 6%; and had a positive effect on lycopene deposition. Tomato pomace also has a positive effect on heat stress in chickens. Hosseini-Vashan et al. [138] showed that feeding 5% TP to chickens at 1–28 days of age can increase body weight and production index, reduce feed conversion rate, serum triglyceride, and HDL cholesterol concentrations, and increase the activity of glutathione peroxidase (GPx) and superoxide dismutase (SOD). However, malondialdehyde (MDA) decreased with low concentrations of TP. Dried TP supplementation did not affect chicken growth performance during heat stress, but it ameliorated the negative effects of heat stress on serum enzyme activity, GPx activity, and lipid peroxidation. This may be related to the carotenoids in TP, because the antioxidant capacity of carotenoids is well known, and antioxidants have a good effect on improving heat stress [139,140,141]. Rezaeipour et al. [142] showed that, in diets supplemented with 5%, 10%, 15%, and 20% TP, although the feed intake increased with an increase in the supplementation level, when the addition amount exceeded 15%, the apparent digestibility of the nutrients and chicken body weight, the apparent metabolizable energy, and the digestibility of crude fat began to decline. This may be because the addition of a level of TP that is too high increases the hardness of the feed and reduces palatability [143]. On the other hand, because the TP is dry, it will expand when it is ingested in the body when it encounters water, thereby increasing its volume in the digestive tract, and produce a feeling of satiety, thus reducing feed intake. Yolao and Yammuen-art [144] reported that the addition of TP increased the average daily gain (ADG) of chickens and had no effect on the feed conversion ratio (FCR), but lowered the heterophile/lymphocyte (H/L) ratio. This suggests that dried TP can reduce the stress response of chickens, but the catalase level in the group fed 20% TP was significantly higher than that in the control group, indicating a greater breakdown of hydrogen peroxide in the body. However, higher levels of TP can be supplemented in ducks. Omar et al. [145] showed that feeding 20% TP significantly increased live weight and feed intake and decreased total cholesterol, triglyceride, and HDL concentration, and its economic benefits are highest when supplemented with 20% TP. In conclusion, when chickens were supplemented with less than 10% TP, the growth performance and antioxidant activity increased with supplementation; when the supplementation level exceeded 15%, the performance showed a downward trend. However, the best level in ducks is 20%, which is slightly higher than the best supplementary level of chickens, which may be related to physique [146,147]. The above results show that supplementing TP in chicken diets can increase feed intake. However, it can also reduce the feed conversion rate, improve antioxidant and immune capacity, and have a good effect on resisting heat stress. The supplementation level should not exceed 15%.

## 5. Nutrition of TP on Swine

The effect of TP on swine is shown in Table 7 Biondi et al. [48] replaced 15% corn with TP in pig diets. The results showed that while TP did not affect its growth performance, meat color, or muscle antioxidant capacity, and decreased the intramuscular fat, SFA, and MUFA content, it increased the PUFA, concentrations of PUFA n-3 and PUFA n-6, and n-6:n-3 ratio. An et al. [148] directly supplemented lycopene and ketchup in pig diets, and the results showed that lycopene and/or ketchup did not affect production traits or plasma lipids, including total lipids, total cholesterol, high-density and low-density cholesterol, and triglycerides, but reduced MDA concentrations in fresh pork belly. Yang et al. [149] also showed that TP does not affect the growth performance and nutrient digestibility of pigs, but did show that it improved antioxidant and biochemical activities. Fachinello et al. [150] found that lycopene supplementation in sow diets increased superoxide dismutase in the liver and, with the increase in lycopene concentration, decreased total cholesterol, LDL, and the LDL:HDL ratio, as well as the gene expression of catalase. Meng et al. [151] showed that lycopene increased serum CAT activity, serum TC concentration, jejunal SOD activity, and the mRNA and protein expression of NRF2 and CD36 in the jejunum of early weaned piglets. It also improved intestinal morphology and increased the villus height, villus/crypt Litter ratio, and abundance of beneficial flora. At the same time, the protein expression of KEAP1 and the abundance of pathogenic flora (such as *Treponema_2* and *Prevotellaceae_unclassified*) decreased. Watanabe et al. [152] added lycopene to an in vitro maturation medium, and the results showed that lycopene delayed the interruption of communication between oocytes and cumulus cells, and increased glutathione levels and fertilization rates in mature oocytes. Wen et al. [153] showed that the dietary supplementation of 200 mg/kg lycopene increased the muscle redness α* value, intramuscular fat, crude protein content, and antioxidant capacity and slowed myosin heavy chain (MyHC) protein levels and muscle fibers. The muscle lightness L* and yellow b* values, fast myosin levels, and percentage of fast twitch fibers decreased. This suggests that lycopene can promote the transition of muscle fiber types from fast-twitch to slow-twitch, while increased a* values of muscle redness may be related to the deposition of lycopene [154]. The above results show that TP or the TP extract lycopene does not affect the growth performance of pigs, but can increase the concentration of PUFA and PUFA n-3 in muscles, improve muscle antioxidants, immune capacity and oocyte fertilization rates, and improve intestinal health; at the same time, lycopene can improve the color, nutritional value, and juiciness of pork.

## 6. Nutrition of TP on Ruminants

The effects of TP on ruminants are shown in Table 8 Valenti et al. [33] showed that the ad libitum consumption of TP did not affect growth performance or lipid oxidation, but increased L*, b*, C*, and H* and decreased muscle 2-thiobarbituric acid reactivity substances (TBARS), indicating that TP has a good antioxidant effect on flesh color and muscle. Abdullahzadeh [155] obtained similar results: supplementing different levels of TP had no effect on goat and sheep body weight, hot carcass, slaughter rate, carcass length, blood sugar, total protein, urea, or cholesterol. However, 30% supplementation increased the content of crude fat and crude protein in muscle, indicating that TP did not affect the growth performance of meat goats but could improve the nutritional value of meat. Moreover, the dietary supplementation of 5%, 10%, and 15% had no effect on the digestibility of dry matter and crude protein in lambs, but the supplementation of 10% and 15% improved the digestibility of OM, CF, EE, and NFE. As supplementation levels increased, FW, TBWG, ADG, and TVFA improved, while the rumen pH and ammonia nitrogen concentration decreased [156]. Abbeddou et al. [157] found that TP reduced the milk production and protein content of goats, but increased the milk fat content and the n-6:n-3 ratio. It had no effect on the ratio of conjugated linoleic acid. Mizael et al. [18] found that feeding different levels of TP to lactating ewes did not affect blood biochemical indicators and thyroid hormones, feed efficiency, or feed conversion ratio, but feeding 60% of the level would reduce the body weight of the ewes. Supplementation by 20% and 40% increased milk quality and fat content, respectively. This indicated that, although TP reduced the body weight of lactating goats, it could improve milk quality and fat content, which may be related to the energy level and fatty acid composition of TP itself. However, there are also inconsistent results. Although supplementation of 30% TP does not affect the apparent digestibility of nutrients or the urinary excretion of total purine derivatives, it also has no effect on the composition of milk or the abundance of total bacteria and methanogens. It also reduces urinary N, ruminal microbial N flux, and NH3-N and CH4 emissions, and increases milk linoleic acid, linolenic acid, and total polyunsaturated fatty acid concentrations [158]. The resulting differences in individual indicators may be related to the environment [104]. Overall, TP had no negative effects on goat growth performance, nutrient digestibility, or immune biochemical indicators, but improved antioxidant capacity as well as muscle and milk fatty acid composition, and it was beneficial in lowering rumen pH and reducing CH4 production. Therefore, TP can be supplemented as feed in sheep diets, but the level should not exceed 40%.

Tuoxunjiang et al. [159] replaced silage corn with 10% silage TP in dairy cows and found that, although milk production and milk composition did not change, increased dry DM intake and digestibility and vitamin concentration in milk increased total cholesterol, high-density lipoprotein cholesterol, serum aspartate aminotransferase concentrations, antioxidant capacity, and immune performance. Similar results were obtained by Zhao et al. [160] using fermented TP instead of soybean meal. TP increased dry matter intake (DMI) and 4% fat-corrected milk. There was no effect on average milk production, feed conversion ratio, milk fat, protein, or total solids. Feed costs reduced and benefits increased. However, Tahmasbi et al. [161] obtained different results. There were no significant differences between the mean daily dry matter and the nutrient intake of dry matter, organic matter, NDF, ADF digestibility, or fecal and rumen pH values in the silage-fed TP group. There were no differences in rumen ammonia nitrogen, but supplementation with 15% had the lowest total blood protein concentration, followed by the 7.5% group. There were differences in daily milk production and the percentages of milk protein and fatty acids. Regardless of the findings based on any study, TP did not negatively affect dry matter intake, production performance, or milk quality in dairy cows, suggesting that TP can be supplemented in dairy cows’ diets, with the highest supplementation level at 15%. In addition, there are few studies on the application of TP in beef cattle. At present, a small amount of literature shows that replacing soybean meal with TP at 3.2%, 8%, and 11% can reduce the final body weight by 2.4%, 3.8%, and 4%, respectively, and food intake decreased linearly [162]. However, the author stated in the publication of the rumen fermentation index that TP did not affect feed intake, but increased rumen pH and ammonia nitrogen concentrations. VFA and rumen bacterial counts remained unchanged, and TP did not affect fiber digestibility [163]. The reasons for the inconsistent data are unknown. Other sources also suggest that the use of TP improves rumen digestion and feed efficiency [164]. There is no negative impact on rumen fermentation at least [165].

## 7. Nutrition of TP on Rabbits

The effect of TP on rabbits is shown in Table 9 Peiretti et al. [166] supplemented the rabbit diet with 3% and 6% TP and found that the polyunsaturated fatty acids of the muscle increased, and the yellow (b*) and color values were in the 6% group. However, it does not affect muscle pH, carcass characteristics, muscle nutrient composition, or the antioxidant status of meat. This may be related to the unsaturated fatty acids in TP. Although TP contains high amounts of carotenoids, unsaturated fatty acids can induce oxidation, so antioxidants have no effect [167]. Hassan et al. [168], with directly fed tomato extracts, improved growth performance, carcass weight, antioxidant status, regulated plasma and meat AA levels in rabbits, while reducing kidney, abdominal, and back fat and meat ether extracts, as well as plasma total cholesterol and low-density lipoprotein cholesterol concentrations, thereby improving economic efficiency. Similar results were obtained when Hassan et al. [169] used 100, 200, and 250 mg/kg of TP extract in rabbit feed. TPE improves growth performance and reduces mortality in rabbits. Catalase and glutathione peroxidase were higher at supplements of 200 and 250 mg, whereas plasma total cholesterol, triglycerides, plasma hydrogen peroxide, and malondialdehyde concentrations increased as dietary TPE levels increased. In addition, TPE supplements improve net revenue and economic benefits. Mennani et al. [170] used TP instead of alfalfa, which increased liver weight in the 60% DTP group and waist weight in the 30% DTP group, while the perirenal fat weight was inversely proportional to the DTP incorporation rate. The 60% DTP group also improved economic efficiency. The addition of low levels of TP had no effect on the blood parameters of the rabbits, but it did improve the final body weight, the feed efficiency values, antioxidant properties, and the immune properties of the rabbits, and the best effect was 2% [171]. Tomato pomace can also promote the reproductive performance of rabbits. Khadr & Abdel-Fattah [172] supplemented with 14%, 22%, and 30% TP, respectively, which had an effect on the average daily feed intake, litter size, and mortality of female rabbits. However, rabbit weaning weight increased. TP had no effect on semen quality, but 30% supplementation improved sperm concentration. This may be related to vitamin E in TP [173]. Vitamin E protects cells from oxidative damage, which maintains sperm quality [174]. Interestingly, El-Ratel [175] directly supplemented 500 mg/kg lycopene in a rabbit’s diet. The result had no effect on the final body weight and water intake, but improved the immune performance of the sperm, reduced lipid content, and improved sperm antioxidant capacity, sperm quantity, sperm quality, and conception rates. In conclusion, TP can improve the growth performance and antioxidant capacity of rabbits, reduce total cholesterol and low-density lipoprotein cholesterol concentrations, and improve semen quality and economic benefits.

## 8. Conclusions and Recommendations 

The rich nutritional value of TP enables its value-added utilization in animal feed. TP can improve animal feed intake and growth performance; increase the PUFA and PUFA n-3 contents in meat; improve meat color, nutritional value, and juiciness; enhance the immunity and antioxidant capacity of animals; and improve sperm quality. Lowering rumen pH and reducing CH_4_ production in ruminants promotes the fermentation of rumen microorganisms and improves economic efficiency. Supplementation levels should not exceed 15% in poultry, 40% in goats, 15% in cattle, and 60% in rabbits to avoid negative effects.

Regarding the value-added utilization of TP, the market economy should lead, and the law should be a guarantee. Only in the case of profit, the reprocessing of TP will change from passive to active, thus generating greater economic value. First, from the perspective of tomato processing production companies, the global tomato processing companies can set up an information sharing platform and formulate a unified standard for the classification and processing of TP, e.g., for cosmetics, medicines, food and feed, and prices, attracting more small and medium-sized tomato processors to join. The platform can then be used to contact such companies in advance before processing, and the obtained orders can be redistributed within the platform according to the principle of the nearest and the most convenient so as to avoid vicious competition and preserve the industrial chain. Second, the platform can set up a low-tech agency processing company (e.g., for feed processing), and the unsalable TP can be transported to this company for processing and sales, which would incur much lower costs than those incurred by tomato processing plants. Finally, the law can ensure the sustainable value-added utilization of TP. This can not only reduce environmental pollution and reduce a company’s cost of processing TP, but also increase the company’s income and promote the development of other industries. In short, as long as it is profitable, TP disposal will no longer be a problem.

## Figures and Tables

**Figure 1 animals-12-03294-f001:**
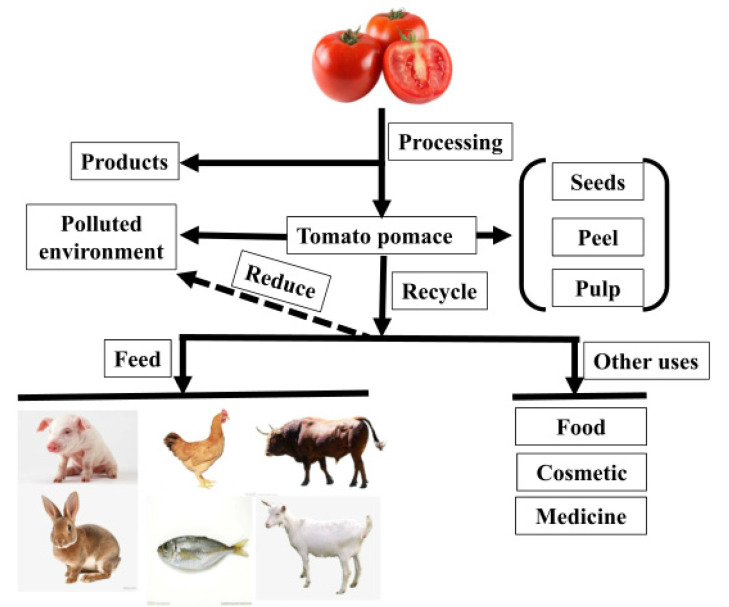
The processing and uses of TP.

**Figure 2 animals-12-03294-f002:**
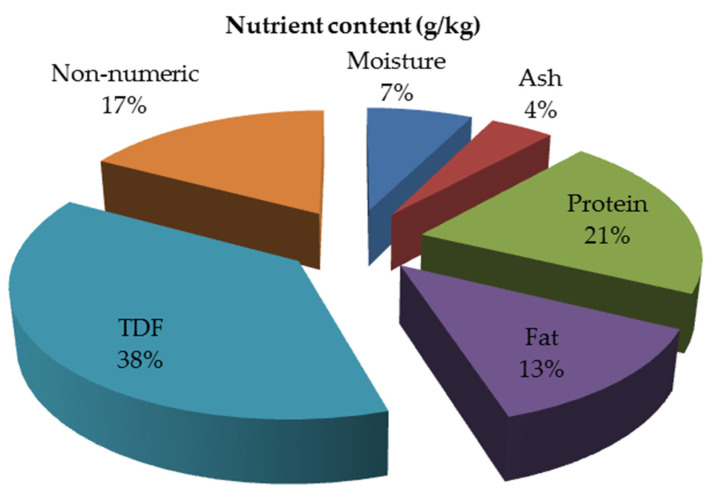
Nutrient content of dried TP (g/kg).

**Figure 3 animals-12-03294-f003:**
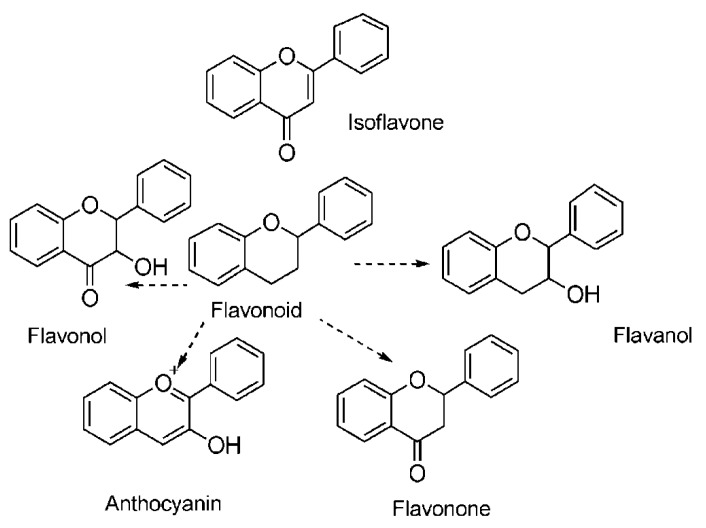
Chemical structure of flavonoids (Reprinted with permission from Ref. [103], 2020, Ullah et al.).

**Table 1 animals-12-03294-t001:** Nutritional content of tomato pomace (g/kg).

Moisture Ash	43.4	79.0 70.1	88.4 27.5	37.0	59.6 42.7	81.0 41.4	59.0 45.0
Protein Fat TDF	298.5 244.7 413.5	201.4 88.3 641.1	186.7 144.5 123.2	219.0 159.0 115.0	149.5 85.2 663.0	173.2 86.5 305.4	217.0 93.0
References	[31]	[32]	[33]	[34]	[35]	[36]	[37]

TDF = total dietary fibers.

**Table 2 animals-12-03294-t002:** The antioxidant content of tomato pomace (g/kg dry matter basis).

TPC (mg GAE/g)	199.4	179.0	94.5	122.9	213.4
TFC (mg QE/g)	102.1	68.77	378.7	41.5	30.6
Lycopene content (g/ kg)	36.7		41.4	50.1	50.2
DPPH radical scavenging activity (%)	52.5	52.4	29.9		75.0
β-carotene bleaching inhibition activity (%)	80.6		149.8	95.6	211.0
References	[32]	[35]	[38]	[39]	[40]

TFC = total flavonoid content; TPC = total phenol content.

**Table 3 animals-12-03294-t003:** Mineral content of tomato pomace (g/kg dry matter basis).

Calcium (Ca)	160.0	76.4	371.5	141.6	131.8	140.5
Phosphorus (P)		219.7				
Magnesium (Mg)	149.0	126.7	3.1	251.1	211.0	157.3
Sodium (Na)	73.6	129.1	191.7	47.2	66.5	78.3
Potassium (K)	1097	1011.5	808.5	667.9	303.0	1125.0
Iron (Fe)	1.5	9.3	11.0		5.6	1.7
Zinc (Zn)	3.12	3.46	1.8		6.3	0.5
References	[41]	[35]	[32]	[42]	[39]	[43]

**Table 4 animals-12-03294-t004:** Fatty acid composition of tomato pomace of different tomato varieties (g/kg dry matter basis).

Source	Amarelo	Caracal, Romania	Waltinger	Red Currant	Batateiro	Comprido
C12:0	1.5				1.1	0.4
C14:0	9.3	4.1	0.9	0.6	6.2	3.2
C15:0	1.5	0.9	0.1	0.2	1.1	0.8
C16:0	205.3	163.2	133.9	133.9	193.1	159.6
C17:0	3.3	1.9	0.9	2.4	2.5	1.8
C18:0	63.4	54.3	43.5	46.7	54.7	63.6
C20:0	12.6		4.8	5.8	8	6.1
C22:0	8.2		1.5	1.8	5.5	3.1
C23:0	15.2		0.2	0.2	7.8	1.6
C24:0	10.1	2.9	1.7	2	7.3	4.5
C25:0			0.3	0.3		
C26:0			0.8	0.9		
SFA	322.2	227.2	190	197.7	289.1	245.7
C16:1	2.5	6.4	2.3	1.5		
C18:1n9	106	185	181.3	198.6		
C20:1	1.2		0.8	1		
C22:1n9	0.3		0.2	0.4		
MUFA	110	197.5	192.4	207.9	130.8	176.6
C18:2n6	398	519.1			463.3	520.5
C20:4n6	0.6				0.4	0.1
C22:2n6		3.9	0.1	0.4		
C22:3n6		5.5				
n-6 PUFA	398.6	530.7			463.7	520.6
C18:3n3	155.3	35.5			114.1	55.5
C20:3n3	2				0.9	0.8
C20:5n3	0.3	2.6			0.5	0.4
C22:3n3		1.3				
n-3 PUFA	157.6	42.2			115.5	56.7
MUFA/SFA, %	34.1	86.9	101.1	105.2	45.2	71.9
n-3 PUFA/n-6 PUFA, %	39.5	8			24.9	10.9
References	[40]	[39]	[44]	[44]		

MUFA = monounsaturated fatty acid; PUFA = polyunsaturated fatty acids; SFA = saturated fatty acid.

**Table 5 animals-12-03294-t005:** Amino acid composition of tomato pomace (g/kg dry matter basis).

Source	Caracal, Romania	Turkey (Seed)	No Message	Cairo, Egypt	Sicily, Italy	Best Factory (Peel)	Pig Requirements	Poultry Requirements
Indispensable amino acids								
Arginine	14.6	10.6	1.8	10.4	10.9	43.4	2.4	10
Histidine		2.6	0.5	4.6	5.1	36.4	2.1	2.7
Isoleucine	6.9	2.9	0.8	9.6	6.3	38.6	3.7	6.2
Leucine	10.7	6.4	1.5	14.6	11.9	50.7	6.7	9.3
Lysine	8.8	5.9	1.7	10.4	7.9	44	6.6	8.5
Methionine	2.7	3.1		1.2	4	10.2	1.8	3.2
Phenylalanine	6.1	9		9.8	7.2	50.2	4	5.6
Threonine	5.5	4.3		8.1	6	23.4	4.3	6.8
Tryptophan					6	34.2	1.2	1.6
Valine	5.4	3.6	1.2	12.3	7	45.8	4.5	7
Dispensable amino acids								
Alanine	7.1	4.7	1	10.7	7.9	50.2		
Aspartic acid	15.7	10.3	2.4	32.9	13.2	7		
Cysteine	2.3	3.1	0.5	2.7	4.1	3.9		
Glycine	6.3			12.7	6.7	42.9		
Glutamic acid	72.1	4.8	5.4	61.2	29.7	145.6		
Proline		4.3	0.9		11.1	27.8		
Serine	1.7	4.5	1	3.7	7.5	30.8		
Tyrosine	6.9		2.5	7.1	2.3	34.2		
Total amino acids	172.4	80.1		131	156	719.3		
References	[39]	[45]	[46]	[47]	[48]	[41]	[49]	[50]

**Table 6 animals-12-03294-t006:** The effect of tomato pomace on poultry.

References	Species/ Breed	Age	Type	Level, %	Performance
[19]	Male IR and Cobb	1–42 days	DTP	4, 6	Increased feed cost, total variable cost, and total cost by 4% and 6%, and feeding TP consumed more feed.
					Lower pH. No negative effect of adding 6% on growth performance parameters, WHC or drip loss, mRNA expression of GHR or IGF-1.
[137]	Japanese Quail	8 weeks	DTP	3, 6, 9, 12	Improves immune performance, antioxidant properties, and digestive enzymes. Lower cholesterol, LDL.
					Increased HDL, egg weight, and hatchability, the largest of which was 6%, had a positive effect on lycopene deposition.
[138]	Male Arian	1–42 days	DTP	3, 5	Increased body weight and production index from 5%. Reduced feed conversion ratio in 5%. Reduced serum triglyceride and HDL cholesterol concentrations on Day 28 from 5%. Increases GPx and SOD activities and decreases MDA from 5%. No effect on growth performance. Improved serum enzyme activity, GPx, and lipid peroxidation during heat stress.
[142]	Ross 308	21–42 days	DTP	5, 10, 15, 20	Decreased body weight in 15% and 20%. Increased feed intake. Decreased nutrient apparent digestibility and crude fat apparent metabolizable energy and apparent digestibility.
[144]	Cobb-500	4–6 weeks	DTP	10, 15, 20	Increased weight gain and ADG. No effect on FCR. Lower heterophil/lymphocyte (H/L) ratio. Increase in catalase level from 20%.
[145]	Wild duck	1–72 days	DTP	10, 15, 20	Increased live weight and feed intake and the most economical from 20%. Decreased total cholesterol, triglycerides, and HDL from 20%. No effect on LDL and total protein.

ADG = average daily gain; DTP = dried tomato pomace; DTP = dry tomato pomace; FCR = feed conversion ratio; GHR = growth hormone receptor gene; GPx = glutathione peroxidase; HDL = high density lipoprotein; IGF-1 = insulin-like growth factor-1; LDL = low density lipoprotein; MDA = malondialdehyde; SOD = superoxide dismutase; WHC = water holding capacity.

**Table 7 animals-12-03294-t007:** The effect of tomato pomace on swine.

References	Breed	Age	Type	Level, %	Performance
[48]	Nero Siciliano	7 months	DTP	15	No effect on growth performance, flesh color, and muscle antioxidant capacity. Decreased intramuscular fat, SFA, and MUFA content. Increased PUFA, PUFA n-3 and PUFA n-6 concentrations, and the n-6:n-3 ratio.
[148]	Landrace × Yorkshire × Duroc	18 weeks	Lycopene (Ly) Ketchup (Kc)	Ly 20 Kc 3.4 Ly 10 + Kc 1.7	No effect on production traits, plasma lipids, including total lipids, total cholesterol, high-density and low-density cholesterol, and triglycerides.
					Decreased MDA concentration in fresh pork belly.
[149]	Barrows	BW: 50.3 ± 1.1 kg	DTP	50 or 100 g/kg	No effect on growth performance and digestibility of nutrients. Increased GSH-Px and glucose, total protein, and globulin.
[150]	Piétrain × Landrace × Large White	BW: 75.04 ± 1.6 kg	Lycopene	12.5, 25.0, 37.5 or 50.0 mg/kg	Decreased SOD in the liver, total cholesterol, LDL, HDL, and LDL:HDL. Decreased catalase gene expression, plasma urea, and triglyceride concentrations.
[151]	Duroc × Landrace × Yorkshire	21–49 days	Lycopene	50 mg/kg	Increased serum CAT activity, TC concentration, and jejunal SOD activity. Decreased serum and jejunal H2O2 concentrations. Increased mRNA and protein expression of NRF2 and CD36 and decreased KEAP1 expression in the jejunum. Increased villus height, villus/crypt ratio, and abundance of beneficial flora; decreased abundance of pathogenic bacteria.
[152]	Sow	Culture medium	Lycopene	10 IU/mL	Delayed disruption of communication between oocytes and cumulus cells. Increased glutathione levels and fertilization rates in mature oocytes.
[153]	Duroc × Landrace × Yorkshire	BW: 63.89 ± 1.15 kg	Lycopene	100 or 200 mg/kg	Increased muscle redness a* value, intramuscular fat, crude protein content, and antioxidant capacity. MyHC protein levels and percentage of slow-twitch fibers at 200 mg/kg. Decreased muscle lightness L* and yellow b* values, fast myosin levels, and percentage of fast-twitch fibers at 200 mg/kg.

CAT = catalase; DTP = dry tomato pomace; GSH-Px = glutathione peroxidase; HDL = high density lipoprotein; LDL = low density lipoprotein; MAD = malondialdehyde; MUFA = Monounsaturated fatty acid; MyHC = slow myosin heavy chain; PUFA = Polyunsaturated fatty acids; SFA = Saturated fatty acid; SOD = superoxide dismutase; TC = total cholesterol.

**Table 8 animals-12-03294-t008:** The effects of tomato pomace on ruminants.

References	Breed	Age	Type	Level, %	Performance
Goat					
[33]	Comisana	45 days	DTP	Feel free to provide	Increased L*, b*, C*, and H*. No effect on growth performance and lipid oxidation.
					Decreased TBARS.
[155]	Markhoz	BW: 18.6 ± 0.7 kg	DTP	10, 20, 30	No effect on body weight, hot carcass, slaughter rate, carcass length, blood sugar, total protein, urea, or cholesterol. Increased crude fat and crude protein content in muscle at 30%.
[156]	Ossimi	BW: 19.25 ± 0.18 kg	DTP	5, 10, 15	No effect on the digestion of DM and CP and total blood lipids. Increased digestibility of OM, CF, EE, and NFE at 10% or 15%. Increased FW, TBWG, ADG, and TVFA. Decreased rumen pH and ammonia nitrogen concentration.
[157]	Awassi	3–6 years	DTP	30	Decreased milk production and milk protein content. Increased milk fat content, n-6:n-3 ratio. No effect on conjugated linoleic acid ratio.
[18]	Saanen	BW: 46.2 ± 7.50 kg	DTP	20, 40, 60	Reduced weight at 60%. Increased milk production and fat mass at 20% and 40%. No effect on feed efficiency and feed conversion ratio, blood glucose, cholesterol, urea, albumin, T3, and T4.
[158]	Murciano-Granadina	BW: 39.4 ± 5.39 kg	DTP	35	No effect on nutrient apparent digestibility, the urinary excretion of total purine derivatives, milk production and composition, or total bacterial and methanogen abundance.
					Decreased N in urine, microbial N flux in rumen, NH3-N and CH_4_.
Cattle					
[159]	Holstein cow	--	ETP	10	No effect on milk yield and composition. Increased vitamin concentration in milk, DM intake, and digestibility. Increased concentrations of total cholesterol, high-density lipoprotein cholesterol, serum aspartate aminotransferase, antioxidants, IgA, IgG, and IgM.
[160]	Xinjiang brown cow	--	FTP	14	Increased DMI and 4% fat-corrected milk. No effect on average milk yield, feed conversion ratio, milk fat, protein, or total solids. Reduced feed costs and increased benefits.
[161]	Holstein cow	BW: 594.2 ± 37.8 kg	ETP	7.5, 15	No effect on dry matter and nutrient intake. No effect on digestibility of dry matter, organic matter, NDF, or ADF. No effect on fecal and rumen pH, or rumen ammonia. No effect on daily milk production, or the percentages of milk protein and fatty acids. Reduced total blood protein.

ADF = acid detergent fiber; ADG = average daily gain; b∗ = yellowness; CF = crude fiber; C* = Chroma; CP = crude protein; DM = dry matter; DMI = dry matter intake; DTP = dry tomato pomace; EE = ether extract; ETP = ensiled tomato pomace; FTP = fermented tomato pomace; FW = final weight; H* = hue; L* = lightness; NDF = neutral detergent fiber; NFE = nitrogen free extract; OM = organic matter; TBARS = 2-thiobarbituric acid reactive substances; TBWG= total body weight gain; TVFA = total volatile fatty acids.

**Table 9 animals-12-03294-t009:** The effect of tomato pomace on rabbits.

References	Breed	Age	Type	Level, %	Performance
[166]	Hycole × Grimaud	38 days	TP	3, 6	No effect on muscle pH, carcass characteristics, muscle nutrient composition, and antioxidant status of meat.
					Increased muscle polyunsaturated fatty acids, and the yellow (b*) and chromatic values at 6%.
[168]	V-Line Male Rabbit	5 weeks	TPE	200 g	Increased SOD activity, economic benefits, growth performance, antioxidant status, regulation of AA levels in plasma and meat, and carcass weight. Reduced plasma total cholesterol and LDL. Reduced fat of the kidney, belly, and back.
[169]	NZW	6 weeks	TPE	100, 200, 250 mg/kg	Heaviest body weight, lowest feed intake, and best feed conversion ratio at 250 mg/kg. Reduced mortality. Increased catalase and glutathione peroxidase. Decreased plasma total protein, globulin, catalase, and glutathione peroxidase. Improved net income and economic benefits.
[170]	Bai Rabbit	33 days	DTP	30, 40, 60	Increased liver weight at 60% and waist weight at 30%. Increased economic benefits. Decreased perirenal fat mass.
[171]	NZW	45 days	DTP	1, 2	Increased final body weight and feed efficiency values. No effect on PCV, Hb, MCV, MCH, MCHC, lymphocytes, monocytes, neutrophils, or eosinophils. Increased phagocytic activity of leukocytes, IgG, IgM, and IgA. Increased serum and liver TAC, SOD, GST, and CAT.
[172]	Mature rabbit	6–8 months	DTP	14, 22, 30	No effect on average daily feed intake, litter size, and mortality rate. Increased weaning weight. No effect on semen color and consistency, pH, sperm motility and viability, total protein, albumin, and globulin in semen. Increased ejaculation volume (at 30%) and sperm cell concentration.
[175]	NWZ male	5 months	Lycopene	500 mg/kg	No effect on FBW and water intake. Increased hemoglobin concentration, hematocrit value, red blood cell, platelet counts, serum total protein, albumin, globulin, glucose, and HDL concentrations. Decreased MAD, white blood cell count, serum urea concentration, creatinine concentration, total lipids, triglycerides, total cholesterol, and LDL concentrations. Increased total antioxidant and testosterone concentrations. Improved sperm quantity, quality, total sperm output, initial semen fructose concentration, and conception rate.

AA = amino acid; b* = yellowness; CAT = catalase; DTP = dry tomato pomace; GST = glutathione transferees; Hb = whole blood hemoglobin concentration; HDL = high density lipoprotein; LDL = low density lipoprotein; MCH = mean corpuscular hemoglobin; MCHC = mean corpuscular hemoglobin concentration; MCV = mean corpuscular volume; MDA = malondialdehyde; PCV = packed cell volume; SOD = superoxide dismutase; TAC = total antioxidant capacity; TPE = tomato pomace extract.

## Data Availability

Not applicable.

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
