# Peer review of "Sustainable Valorization of Tomato Pomace (Lycopersicon esculentum) in Animal Nutrition: A Review"

_animals, 2022, doi:10.3390/ani12233294_

Round 1

Reviewer 1 Report

The Ms by Lu et al. entitled “  Sustainable Valorisation of Tomato Pomace (Lycopersicon es-2 culentum) in Animal Nutrition: A Review”  reviewed  the  background of the current global challenge  of feed resources, especially the shortage of protein feed, human beings are constantly trying to develop and utilize new feed resources. The MS summarizes the nutritional value of TP and its use and impact in animals as an animal feed supplement. TP is a by-product of tomato processing, divided into peel, pulp and tomato seeds, which rich in proteins, fats, minerals, fatty acids and amino acids, as well as antioxidant bioactive compounds such as lycopene, beta-carotenoids, tocopherols, polyphenols and terpenes. As feed, there are mainly two forms of drying and silage. The authors concluded that TP can enhance animal feed intake, growth performance, and increase PUFA and PUFA n-3 content in meat, improve meat color, nutritional value and juiciness, enhance immunity and antioxidant capacity of animals’ tissue. In addition, enrich improve sperm quality, decreasing rumen pH and lowering CH4 production in ruminants promotes fermentation of rumen microorganisms and improves eco-nomic efficiency.  The Tomato Pomace instead of soybean meal as a protein supplement is a research hotspot in the animal industry, and further research should focus on the processing technology of TP and its large-scale application in feed.  The MS discuss a good topic after changes in feed chain due to COVID-19 and current war between R-U. There are several comments that would help the authors improve the outcomes of this MS, for example:

1.       The simple summary is lacking?

2.      L 25, please take care of the English editing rules   , Improve ?

3.       It is not good to starter the sentence with the abbreviation, i.e. TP

4.      The novelty of the work has to be added to  the introduction section, please emphasis on the added value/novelty of this good review article as there are some works that have been done.

5.      Plz update references with, L 39, as references 5 and adjust references number thereafter

- Md. Tanvir Rahman, Md. Saiful Islam, Shehata AA, Shereen Basiouni, Hafez M Hafez, Esam I. Azhar, Asmaa F. Khafaga, Fulvia Bovera, Youssef Attia, (2022). Influence of COVID-19 on the sustainability of livestock performance and welfare on a global scale. Tropical Animal health and Production 2022,  DOI:10.1007/s11250-022-03256

6.  Please in M&M section add the method of collecting the literature used, and how you dealt with.

7. Please fix the column width

8.  L 97, Table 1. Nutritional content of Tomato Pomace (g/kg).

9. The title of figure 2, showed be corrected as Nutrient content of dried TP (g/kg)

10. In the all tables heading, don’t use the abbreviation, table are independent content of the MS.

11. L 135, Table 4. Fatty acid composition of TP (g/kg dry matter basis) of different Tomato varieties, correction is needed

12. L 140, to50.2 g/kg, space is needed

13. The amino acids composition should be compared with nonruminants animals requirements such as NRC, 1994.

14. The head of tables and head of the column needs more indications, i.e. Table 5  Amino acid composition of TP (g/kg dry matter basis).

15. Authors, should focus on essential amino acids and essential fatty acids in the discussion and presentation of the data.

16. Antioxidant mechanisms of bioactive substances should be better presented in diagram.

17. We conclude, The review indicates  that ……………., Plz replace

18.  In conclusion section, I personally believe that, Plz delete

19.  I recommend a major revision and English editing 

Author Response

Dear Reviewer,

    Thank you very much  taking your precious time out of your buzy schedule to give valuable suggestions to this veriew. I have completely revised it according to your suggestions. The revision instructions are in attachment, please refer to it.

     Bset Regards,

     Student, Lu Shengyong.

Reviewer 2 Report

Dear authors,

Thank you for submitting this paper that investigates the use of tomato pomace as a supplement for animal diets. You have made some good arguments for sustainability. This work is very well researched.

At current however, there seem to be some large revisions required in the manuscript to ensure the work is scientifically robust. I have attached the PDF version of the manuscript with specific comments. Additionally, please consider the following points: 

1. Length of work. Currently, the work is excessively lengthy and the wider message is lost as a result. Reduce the length of the text so the key message is clearer.

2. Tables. Currently, some of the tables are unreadable. Please make sure the tables are reformatted.

With these revisions, the paper should be in a better position overall.

Author Response

Dear Reviewer,

     Thank you very much for taking your precious time out of your busy schedule to give valuable suggestions to this review. I hvae completely revised it according to your suggestions. The revision instructions are in the attechment, pleaserefer to it.

     Best Regards,

    Thank you very much.

     Student, Lu Shengyong

Round 2

Reviewer 1 Report

I have no further comments 

Reviewer 2 Report

Dear Authors,

Many thanks for submitting this revised version of the manuscript for review. You have taken into account the feedback provided on the initial review of the paper. You have also shown clearly where changes have been made to the work, as shown with the highlighted sections of text. The developments to the manuscript have resulted in a more robust paper overall. In light of the revisions, the paper is now in a much better position for consideration.